Automated guided vehicle (AGV) path optimization method based on improved rapidly-exploring random trees

Ren Zhigang 1 2
Cai Anjiang 1 cai_aj12315@outlook.com
Xu Feilong 2
1 School of Mechanical and Electrical Engineering, Xi’an University of Architecture and Technology , Xi’an , China
2 Xi’an Huayun Wisdom Information Technology Co., Ltd. , Xi’an , China
Shah Syed Hassan
Electronic publication date: 2025 Jun 18
Publication date: 2025
Volume: 11
Electronic Location ID: e2915
Received 2025 Mar 18; Accepted 2025 May 5
Copyright: © 2025 Ren et al.
Copyright year: 2025
Copyright holder: Ren et al.
License: This is an open access article distributed under the terms of the Creative Commons Attribution License, which permits unrestricted use, distribution, reproduction and adaptation in any medium and for any purpose provided that it is properly attributed. For attribution, the original author(s), title, publication source (PeerJ Computer Science) and either DOI or URL of the article must be cited.
License URL: https://creativecommons.org/licenses/by/4.0/

Keywords: Path planning, Adaptive step size, Rapidly-exploring random tree, Fast nearest neighbor search

Funding: The authors received no funding for this work.

==============================
In response to the issues of low computational efficiency, slow convergence speed, curvy paths, and the tendency to fall into local optima in rapidly-exploring random tree (RRT) algorithms for automated guided vehicle (AGV) path planning, this article proposes an improved RRT algorithm that combines adaptive step-size optimization with K-dimensional tree (KD-Tree) based fast nearest neighbor search. Firstly, an adaptive step-size optimization strategy is introduced to dynamically adjust the step size during node searches, improving both the planning quality and computational efficiency of the algorithm. Secondly, the KD-Tree nearest neighbor search method is employed to accelerate node searching and reduce the time cost of path planning. Finally, a cubic spline interpolation function is applied to smooth the optimal path, further enhancing the planning quality. Experimental results show that the improved RRT algorithm significantly outperforms traditional RRT, RRT*, and Informed-RRT* in terms of path length, planning time, and path smoothness. Specifically, the average path length is reduced by 164.33 m, and the average search time is shortened by 3.3 s, making it more suitable for AGV path planning in practical applications.

Introduction

With the continuous advancement of industrial automation, automated guided vehicle (AGVs), as unmanned transport vehicles, are capable of autonomous navigation and operation from start to finish through preset paths or by employing simultaneous localization and mapping (SLAM) technology. This not only reduces production costs and enhances efficiency but also effectively mitigates safety risks in the work environment. AGVs have found extensive applications in various fields such as logistics and transportation (Vaca-Recalde et al., 2024), industrial manufacturing (Xin et al., 2024), and intelligent warehousing (Sun et al., 2023).

Path planning technology (Bozzi et al., 2025; Zhang et al., 2024), as the cornerstone of AGV’s autonomous navigation, directly impacts the safety and efficiency of AGV operations through its timeliness and accuracy (Li et al., 2025; Rajaram & Baskar, 2025; Dvorak et al., 2024). In current path planning research, commonly used benchmark algorithms include Dijkstra’s algorithm (Mo, Yu & Du, 2024), A* algorithm (Wu, Chen & Liu, 2024), ant colony algorithm (Hou et al., 2024), and the rapidly-exploring random tree (RRT) algorithm (Kim, Ahn & Park, 2024), among others. Specifically, Dijkstra’s algorithm identifies the shortest path by selecting the vertex closest to the starting point from a set of vertices with undetermined shortest paths, moving it into the determined set, and updating the distances of adjacent vertices to the starting point. However, this method’s high computational complexity significantly increases computation time when dealing with large-scale maps, presenting obvious limitations in scene requiring high real-time performance. The A* algorithm improves search efficiency by introducing a heuristic function to guide the search direction, yet designing an effective heuristic function in complex environments can be challenging, and it lacks flexibility in dynamic planning. The ant colony algorithm boasts good global search capabilities and robustness, but suffers from slow search speeds due to initial pheromone scarcity and a tendency to fall into local optima. The RRT algorithm, known for its flexibility and adaptability to high-dimensional spaces, simulates the growth process of a tree, starting from the initial point, randomly selecting state points in the search space to gradually build a tree structure, continuously exploring the space until it covers the target area, ultimately forming an effective path from start to goal. It is particularly adept at quickly exploring the entire search space in narrow paths, making it widely applicable in AGV path planning (Yu et al., 2024). However, the paths generated by the RRT algorithm can be somewhat random and may not be optimal, with pathfinding efficiency limited in environments with large-scale dense obstacles, requiring numerous iterations to successfully find a path (Zhang et al., 2020).

In order to accelerate the path search efficiency and improve the path planning quality, different researchers have made various improvements and optimizations to the RRT algorithm from different perspectives. For example, the RRT* algorithm (Wang et al., 2024) dynamically updates the connecting nodes through resampling and optimality guarantee strategies, effectively improving both path optimization quality and efficiency. However, the exploration speed in the early stages of the search is relatively slow, and the convergence speed is limited when finding the optimal path in high-dimensional spaces. To address these issues, the Informed RRT* algorithm (Dai, Zhang & Deng, 2024; Miao et al., 2025), based on RRT*, utilizes information about the currently known optimal path to guide the search. By progressively narrowing the search space, it concentrates resources on regions more likely to contain better paths, thus finding the optimal path in a shorter time. However, the algorithm still uses a fixed step-size optimization strategy, and in scenarios where fine adjustments are needed to avoid obstacles, the fixed step size may limit the search speed. Additionally, the planned path lacks smoothness, which affects the operational efficiency of AGVs. Therefore, it is necessary to explore a fast search method that combines dynamic step-size optimization strategies with path smoothing techniques to improve both the quality and speed of path planning. Although RRT and its improved versions perform excellently in many path planning problems, especially in high-dimensional space, existing algorithms still have some significant limitations. In environments with complex obstacle distributions in high-dimensional spaces, RRT* and Informed-RRT propose strategies to optimize the path, but they still cannot effectively balance exploration efficiency and optimization accuracy, resulting in excessively high computational costs in practical applications. Furthermore, existing versions of RRT typically assume that the environment is static, but in many practical applications, the environment is dynamic. These algorithms are not well-suited for dynamic environments and struggle to adjust paths in time to avoid collisions or re-plan paths. Moreover, the step size selection relies on fixed step sizes and simple approximate optimization strategies, which limits the algorithm’s ability to adapt to obstacles of different scales and irregular terrains in complex environments. The generated paths are often curvy, lacking smoothness, which affects both the feasibility and operational practicability of the paths in real-world applications.

To address the above-mentioned issues and improve the path planning quality and efficiency of the traditional RRT algorithm, this article proposes an improved RRT algorithm that combines an adaptive step-size optimization strategy with the K-dimentsional tree (KD-Tree) nearest neighbor fast search. This approach significantly enhances the quality, smoothness, and safety of the path while improving path planning efficiency. The adaptive step-size optimization strategy dynamically adjusts the step size based on the complexity of the environment, optimizing both the speed and accuracy of the path planning. In open areas, larger step sizes are used to accelerate the search, while in areas near obstacles or in complex regions, the step size is automatically reduced to improve path accuracy and avoid collisions. This balances efficiency and safety, enhancing the algorithm’s adaptability in different environments. Additionally, by introducing KD-Tree nearest neighbor fast search technology and efficiently dividing the data structure, the algorithm accelerates the search for the nearest neighbor nodes during node search, enabling faster path searching and further improving the overall efficiency of path planning. Finally, cubic spline interpolation smoothing is applied as a post-processing step to smooth the path generated by the RRT algorithm, eliminating sharp turns and discontinuities in the path, thereby reducing the vibrations and instability that may occur during vehicle movement. The collaborative effect of these three components not only improves the path search efficiency but also optimizes the quality of the path, making the improved RRT algorithm more suitable for AGV path planning in complex environments. The main contributions of the improved rapid-exploring random tree-based AGV path optimization method can be summarized as follows: 1. An optimized path planning algorithm combining an adaptive step-size optimization strategy and nearest neighbor fast search is proposed. The adaptive step-size optimization strategy intelligently adjusts the search step size based on environmental changes. In open areas, it effectively increases the speed of path planning, while in complex environments, it ensures path planning accuracy. This significantly improves the limitations of the traditional RRT algorithm in terms of step size and search efficiency.

2. A new step-size state feedback mechanism is designed, enabling adaptive adjustment of the step size in dynamic environments. Additionally, a KD-Tree data structure is used to establish a fast nearest neighbor search mechanism, allowing the path planning algorithm to precisely locate the nearest neighbor nodes at high speed during the process of building the random tree. This greatly reduces search time and further enhances the overall search efficiency of the planning algorithm.

3. During the path planning process, a cubic spline interpolation function is used to smooth the optimal path, improving its feasibility and safety. By combining this with the adaptive step-size strategy and KD-Tree fast search mechanism, the obstacle avoidance and safety of the path are further enhanced in complex environments. This reduces vibrations and energy consumption during the AGV’s movement. Additionally, the smoothed path is closer to the actual travel path, improving the algorithm’s practicality and reliability.

The structure of this article is as follows: First, the research background, problem statement, and relevant research progress are introduced. Second, in the literature review, it discusses the path optimization strategy of the improved RRT algorithm (Principles of the RRT Path Optimization Algorithm), the conversion relationship between AGV operating speed and search step size (Conversion Relationship Between AGV Operating Speed and Search Step Size), and related research on AGV path planning algorithms (Research on AGV Path Planning Algorithms). In the proposed method section, the path optimization strategy of the improved RRT algorithm (Path Optimization Strategy for the Improved RRT Algorithm), adaptive step size optimization strategy (Adaptive Step Size Optimization Strategy), nearest neighbor fast search and node expansion mechanism (Nearest Neighbor Fast Search and Node Expansion Mechanism), and cubic spline path smoothing processing (Cubic Spline Path Smoothing Processing) are described in detail. In the experimental results and analysis section, a detailed explanation is provided for the experimental setup (Experimental Setup), result analysis and convergence analysis (Result Analysis and Convergence Analysis), and significance analysis (Significance Analysis). Additionally, ablation studies are conducted to explore the impact of hyperparameters (Experimental Study on Hyperparameters), sensitivity to the maximum number of iterations (Sensitivity Analysis of the Maximum Number of Iterations), effects of module removal (Ablation Study on the Removal of Algorithm Modules), and practical applications (Discussion on Practical Applications). Finally, the full text is summarized and future research directions are outlined.

Related research

This section mainly summarizes the basic principles and mathematical model of the RRT algorithm in AGV path planning, as well as the conversion relationship between the AGV’s operating speed and the search step size, along with the current state of research on existing path planning algorithms. This provides the foundation for the proposed improved algorithm.

Principles of the RRT path optimization algorithm

The RRT algorithm (Zhang et al., 2025) is a tree-structured path optimization model. It begins by identifying the start and goal points, creating an initial tree structure with the start point as the root node. Subsequently, it randomly samples new points within the entire search space. By calculating the Euclidean distance between the newly sampled point and each node in the existing tree structure, the algorithm identifies the nearest node to the new point. Starting from this nearest node, it extends a certain step size in the direction of the new sample point to generate a new node. The algorithm then checks whether the new node lies within an obstacle region; if it does, the node is discarded, and the sampling and extension process is repeated. This process of random sampling and extension continues, with the tree structure growing and spreading to gradually explore the entire space. After each extension, the algorithm checks whether the newly generated node is near the goal point or can directly connect to it. If the condition is met, a path from the start point to the goal point is found. The overall process is illustrated in Fig. 1.

Figure 1 Schematic diagram of RRT path node search.

The expansion principle of nodes during the path node search process can be expressed as:

(1) v=(xrand−xnear,yrand−ynear)

(2) v^=v∥v∥=xrand−xnear,yrand−ynear(xrand−xnear)2+(yrand−ynear)2

(3) v=∇v^

(4) xnew=xnear+vnew=xnear+∇

where v represents the vector pointing from the nearest node to the random point, and the unit vector v is obtained by normalizing v^. Starting with the initial point xinit as the root node of the path, a random sample point xrand is generated within the search space. xnear is the nearest child node to this random point. A new child node xnew is expanded along the line connecting xnear and xrand with a fixed step size. If xnew does not collide with any obstacles, it is added to the path nodes of the rapidly exploring random tree; otherwise, the point is discarded, and xnew is regenerated. The above steps are repeated until xnew reaches the goal point or its vicinity, thereby finding a planned path from xinit to the goal point.

Conversion relationship between AGV operating speed and search step size

The step size in the RRT algorithm represents the straight-line distance between two nodes in the path, which corresponds to the distance traveled by the AGV between two steering actions in its driving path. This distance is determined by two factors: the action time of the AGV’s automatic control and the vehicle’s driving speed. According to the technical requirements of AGVs, the minimum action frequency for the autonomous steering control is 2 Hz. The distance traveled by the vehicle between two steering actions is defined as the step size for vehicle path planning, which can be expressed as the product of the vehicle speed and the steering time interval (Mao, Lv & Quan, 2025):

(5) r=u∙t

where u represents the AGV’s motion speed, and t denotes the time interval between steering actions; r is the actual step size at different speeds. When running the RRT algorithm, the actual step size is converted into a pixel step size by incorporating the scale of pixels in the image.

Research on AGV path planning algorithms

In the field of AGV path planning, researchers have proposed various algorithms to improve the navigation capabilities of AGVs in complex environments. These algorithms can be broadly categorized into traditional path planning methods, optimization-based path planning methods, learning-based path planning methods, and hybrid path planning methods. Traditional path planning methods mainly include algorithms such as A*, Dijkstra, and RRT. These algorithms are based on graph search, topological structures, or tree structures and can effectively solve path planning problems in static environments. For instance, the A* algorithm finds the shortest path from the start to the goal through heuristic search and is widely used in AGV path planning; Dijkstra’s algorithm is suitable for finding the shortest path in complex networks and is commonly used in more dynamic environments. However, these traditional algorithms have limitations in handling dynamic environments and real-time obstacle avoidance, as they have relatively high computational complexity and are prone to “curse of dimensionality” in high-dimensional spaces. Optimization-based path planning methods, which incorporate particle swarm optimization, genetic algorithms, and other techniques, optimize AGV path planning by constructing optimization objective functions that comprehensively consider factors such as the shortest distance, smoothness, and obstacle avoidance. These optimization algorithms perform well when dealing with multi-objective and multi-constraint problems, but they typically require high computational costs and may not be able to respond in real-time in dynamic environments. Path planning methods based on deep learning and reinforcement learning, with their adaptability and learning capabilities, can better cope with changes in dynamic environments. However, issues such as long training times and large data requirements remain bottlenecks limiting their widespread application.

The RRT algorithm and its improved versions have been widely applied to path planning problems, especially in complex environments. The RRT algorithm can quickly find a feasible path in a large space by rapidly expanding the tree, making it particularly suitable for high-dimensional space path planning. RRT uses a randomized method to expand the tree structure, efficiently exploring the space and finding feasible paths, which is very suitable for path planning in dynamic environments. However, the paths generated by the traditional RRT algorithm are often rough, lack smoothness, and have limitations in computational efficiency and dynamic obstacle avoidance capability (Caccavale & Finzi, 2022). To address these issues, researchers have proposed various improved algorithms, such as RRT*, Density Gradient-RRT (Huang, Fan & Sun, 2024), Quick-RRT* (Cui et al., 2024), Bi-directional RRT (Sheng et al., 2025), and the Convex Decomposition-based Real-Time Optimal Path Planning algorithm (CDRT-RRT*) (Liu et al., 2025). These improved algorithms optimize path quality, reduce path roughness, increase path expansion speed, and enhance obstacle avoidance capabilities. Among them, RRT* achieves asymptotic optimality of the path by optimizing the cost function of the path; Quick-RRT accelerates the tree expansion speed, thus improving the algorithm’s computational efficiency; Bi-directional RRT expands from both the start and goal simultaneously, reducing path planning time; Density Gradient-RRT considers density information during tree expansion, further improving path quality. CDRT-RRT* improves computational efficiency by dividing the environment into simpler convex regions through convex segmentation, reducing the complexity of path planning, and effectively avoiding obstacles in dynamic environments. However, there are still limitations when dealing with real-time obstacle avoidance, path smoothness, and expansion efficiency in high-dimensional spaces. In environments with significant changes, the convex segmentation method may incur additional computational overhead, thus increasing the time required for path updates. Therefore, future research will focus more on the intelligence and adaptability of algorithms to meet the increasingly complex path planning requirements.

Proposed method

Path optimization strategy for the improved RRT algorithm

This article ingeniously integrates the adaptive step size optimization strategy, the KD-Tree fast search mechanism, and the cubic spline function path optimization. The adaptive step size optimization strategy dynamically adjusts the step size based on different scene: it reduces the step size when approaching the target point to improve precision and performs fine adjustments when encountering obstacles to prevent new nodes from crossing them, thereby significantly enhancing the accuracy of path planning. The KD-Tree fast search mechanism accelerates the search speed for the nearest node, markedly improving the algorithm’s performance and enabling more efficient path planning in complex environments. The cubic spline path optimization smooths the generated path, making it more aligned with practical application requirements and further enhancing the quality of path planning. Through the organic combination of these three techniques, the efficiency and quality of path planning in complex environments can be effectively improved. The improved RRT algorithm process is shown in Fig. 2, and its corresponding pseudocode is presented in Algorithm 1.

Figure 2 Flowchart of the improved RRT path planning algorithm.

Algorithm 1 Pseudocode representation of the improved RRT path planning algorithm.

 1:  Step 1. Initialize start and goal points: start,goal	
 2:  Step 2. Initialize tree with start point: T←{start}	
 3:  Step 3. Set parameters: max_step_size,min_step_size,obstacle_check_resolution	
 4:  Step 4.	
 5:  while tree has not reached goal do	
 6:     a. Sample random point: random_point←RandomSample()	
 7:     b. Find nearest node: nearest_node←KD−TreeSearch(T,random_point)	
 8:     c. Calculate adaptive step size:	
 9:     if distance to goal is small then	
10:         step_size←min_step_size	
11:     else if obstacle detected then	
12:        Adjust step_size to avoid obstacle	
13:     else	
14:      step_size←max_step_size	
15:     end if	
16:     d. Generate new node: new_node←Extend(nearest_node,random_point,step_size)	
17:     e.	
18:     if new_node is valid then	
19:         T.add(new_node)	
20:     end if	
21:     f.	
22:     if new_node is near goal then	
23:       Attempt direct connection to goal	
24:     end if	
25:  end while	
26:  Step 5. Optimize path: optimized_path←CubicSpline(T)	
27:  Step 6. return optimized_path	

Adaptive step size optimization strategy

In the traditional fixed-step RRT algorithm, the same step size is used regardless of environmental changes, which can lead to slow searches and wasted computational resources in open areas. The adaptive step size can increase the search stride in such cases, quickly covering more areas, improving search efficiency, and reducing computation time. During AGV path planning, when approaching the target or navigating complex terrain, a fixed step size may cause the algorithm to miss the target, making it difficult to accurately find a feasible path. This article adopts an adaptive step size optimization strategy, which can reduce the step size for fine-grained searches, allowing the path to more precisely approach the target and increasing the likelihood of the algorithm model finding the optimal path (Wang et al., 2023). First, the algorithm continuously monitors the distance between the current node and the target node. When the distance falls below a specific threshold, the adaptive adjustment mechanism is triggered, and the system enters a fine operation mode. The distance between nodes is accurately measured using the Euclidean distance formula, which can be expressed as:

(6) d(A,B)=(x2−x1)2+(y2−y1)2

where d(A,B) represents the distance between two nodes, where A and B are two nodes in a two-dimensional search space with coordinates (x1,y1), (x2,y2). The step size is dynamically adjusted based on the distance between nodes during the search. In areas closer to the target or with more complex environments, the step size is reduced to improve search precision. In areas farther from the target or with simpler environments, the step size is increased to accelerate the search speed. This approach effectively enhances the planning quality and computational efficiency of the algorithm.

Assume the current node is (x1,y1), and the set of coordinates for the target node and obstacles is C. The minimum Euclidean distance between the current node and the set C is calculated as d. The step size update strategy is as follows:

(7) ∇=∇min+d−dmindmax−dmin(∇max−∇min)

where dmin and dmax are distance thresholds, and ∇min and ∇max are step size thresholds. When d≤dmin, the search is conducted with a smaller step size, and when d>dmax, the search is conducted with a larger step size. In AGV path planning, the distance threshold and search step size are adjusted based on the geometric characteristics of the target area. Specifically, dmax represents the AGV’s maximum travel range, which is set based on the longest straight-line distance to the target area. dmin is determined according to the minimum side length or radius of the target area, typically ranging from 0.2 to 0.4 times the minimum side length or radius, ensuring that the AGV can accurately approach the target by reducing the search step size when it gets closer. The minimum search step size is also set within the range of 0.2 to 0.4 times the minimum side length or radius of the target area, ensuring fine adjustments. The maximum search step size is determined by the AGV’s maximum search speed, ensuring efficient movement during path planning.

Nearest neighbor fast search and node expansion mechanism

KD-Tree (Guo et al., 2023; Sulistianto, Pratama & Widyawan, 2024) is a data structure based on binary trees, commonly used for data storage and search in high-dimensional spaces. In path planning, each node represents a position in space, and the node’s attributes include position coordinates and possibly other relevant information. When constructing a KD-Tree, the data is first preprocessed by collecting spatial position data points, removing outliers and erroneous data, and normalizing the data. Next, the dimension with the largest data variance is selected for partitioning, typically choosing the median of the data to split it into two parts. The process is then repeated on each part to construct subtrees until the stopping conditions are met. In nearest neighbor search, starting from the root node, the search subtree is determined by comparing the target node’s coordinates with the current node’s partitioning dimension. The search continues until a leaf node is reached or the stopping condition is met, at which point the nearest neighbor is recorded, and backtracking occurs to check for closer neighbors. The process involves calculating the distance from the target node to the hyperplane dividing the current node, and continuously updating the nearest neighbor during the search. Finally, based on the nearest neighbor and the information from the backtracking process, a path from the start position to the target position is constructed. This path can be refined by progressively connecting the nodes and optimizing the initial path, checking for collisions with obstacles, and making adjustments, resulting in an optimized path output.

Compared to traditional point-to-point search methods, KD-Tree can quickly narrow down the search range by using spatial partitioning and hierarchical organization, rapidly locating the nearest neighbor node, thereby reducing search time. This is especially advantageous in large-scale node cases and is effective in handling high-dimensional data, adapting to different dimensional space environments. When dealing with dynamic environments, path planning not only relies on 2D position coordinates but also involves multiple dimensions such as speed, acceleration, and load, which collectively determine the motion state of the AGV. As an efficient data structure, KD-Tree can real-time obtain the position and status of obstacles and adjust the path accordingly, helping the AGV avoid obstacles in real-time. In terms of computational complexity, traditional point-to-point search methods have a complexity of O(n). The construction of KD-Tree requires sorting the data and selecting the dimension with the largest variance for partitioning. The time complexity of each partitioning step is O(nlogn), and the nearest neighbor search is used to find the optimal path. In the worst case, the computational complexity of KD-Tree is O(n+logn), and the average computational complexity is O(logn), which is significantly better than traditional path planning algorithms. This efficient search capability makes KD-Tree suitable for high real-time and high-efficiency dynamic path planning, ensuring that the AGV can complete tasks quickly and safely in complex environments.

The construction process of a KD-Tree mainly consists of the following three steps: (1) Determine the split dimension: For all K-dimensional data, calculate the variance of each data point in each dimension respectively. Select the dimension corresponding to the maximum variance value as the value of the split dimension. The larger the variance, the more scattered the data is in the direction of that coordinate axis, and the higher the resolution of the data. (2) Determine the root node: Sort the K-dimensional data according to the value of the split dimension, and the median data point is the root node of the KD-Tree. (3) Determine the left and right subspaces: After determining the root node, the data on the left side of the root node forms the left subtree, and the data on the right side forms the right subtree. Then, select the split dimensions for the K dimensions in turn to build the split dimensions of each layer of the subtree. Subsequently, repeat steps (2) and (3) for the left subtree and the right subtree respectively until there is only the last node left in each subspace. The overall algorithm flow is shown in Fig. 3.

Figure 3 KD-Tree construction algorithm flowchart.

Cubic spline path smoothing processing

The cubic spline function (Lu & Chen, 2024; Li et al., 2024) is a piecewise cubic polynomial function commonly used to construct smooth curves between a series of discrete points. In the path smoothing process, assuming the ordered path points are (xi,yi) (where i=0,1,...,n), to ensure smooth connections between the path points, a function y=s(x) is required such that it is a cubic polynomial in each interval [xi,xi+1]. The cubic spline function s(x) in each interval [xi,xi+1] can be expressed as:

(8) s(x)=ai(x−xi)3+bi(x−xi)2+ci(x−xi)+di

where ai, bi, ci, and di are coefficients to be determined. The cubic spline interpolation function must satisfy certain continuity and smoothness conditions at these points, including continuity of the function value, first derivative, and second derivative. At the node xi, the function values on the left and right sides must be equal, ensuring that the path does not exhibit jumps when transitioning from one interval to another. For two adjacent nodes xi and xi+1, the smoothness constraints can be expressed as:

(9) s(xi)=s(xi+1)

(10) s′(xi−)=s′(xi+)

(11) s″(xi−)=s″(xi+)

where the first derivative constraint ensures smoothness in the tangential direction of the path, while the second derivative constraint ensures smoothness in the curvature of the path. To uniquely determine the coefficients of the cubic spline function, boundary conditions for solving the coefficients must also be defined. Common boundary conditions include natural boundary conditions and fixed boundary conditions. Natural boundary conditions are suitable for cases where there are no special constraints at the ends of the path, allowing for natural curvature. Fixed boundary conditions are used when the starting and ending directions of the path are known. In this article, to make AGV path planning more flexible, natural boundary conditions are adopted as the boundary constraints for solving the coefficients, i.e., s″(x0)=0 and s″(xn)=0.

By applying the aforementioned continuous smoothness constraints and boundary constraints, a system of linear equations for the coefficients of the cubic spline interpolation function can be established. Solving this system of linear equations yields the coefficients corresponding to the interpolation function in each sub-interval, which enables the calculation of path points within these intervals and thereby constructs a smooth path. Implementing cubic spline interpolation for smoothing the planned optimal path can refine the original discrete path into a more continuous trajectory, further enhancing the quality of path planning.

Experiments and result analysis

Experimental setup

To validate the performance of the improved algorithm in path planning, this article conducts comparative experiments between the enhanced RRT algorithm and traditional RRT, RRT*, and Informed-RRT* algorithms. Under the same experimental conditions, path planning is performed using maps of varying scales, as shown in Fig. 4. The three test scene all feature maps with dimensions of 800m×800m. Scenario 1 represents a factory area with cylindrical obstacles that are sparsely distributed, designed to test the algorithm’s decision-making capabilities when faced with multiple path choices. Scenario 2 simulates a warehouse with neatly arranged shelf obstacles, replicating path planning in a structured goods storage space. Scenario 3 involves a complex goods placement area with uniformly arranged but variably sized obstacles, further simulating path planning in narrow passages. The performance of the path planning is evaluated based on two metrics: path length and planning time. The experiments are conducted on a computer equipped with an Intel® Core™ i9-14900HX 2.20 GHz processor, 32 GB RAM, an NVIDIA RTX 4070 Ti GPU, and MATLAB 2024 (The MathWorks, Natick, MA, USA).

Figure 4 Schematic diagrams of map distribution in different scene.

Result analysis

Figure 5 presents a visual comparison of the planning results of the four path-planning algorithms across different scenario maps. The comparison demonstrates that the improved algorithm proposed in this article achieves superior planning performance in all three test scene. It effectively reduces unnecessary exploration in irrelevant regions while delivering smoother path transitions at turning points.

Figure 5 Comparison of path planning results in different test scene.

Using average search time and average path length as evaluation metrics, a comparison of the different algorithms is presented in Table 1. The results demonstrate that the improved RRT algorithm achieves significant enhancements in both path length and search time. Compared to the traditional RRT, RRT*, and Informed-RRT* algorithms, the average path length is reduced by 164.33 units, and the average search time is shortened by 3.39 s. In the improved algorithm, the incorporation of an adaptive step size optimization strategy and a nearest-neighbor fast search method effectively reduces the randomness of the planning process. This results in more stable planning outcomes, better meeting the path-planning requirements of AGVs in practical applications.

Table 1 Comparison of path planning parameters of different algorithms.

Testing scene	Evaluation method	RRT	RRT*	Informed-RRT*	Ours	
Scene1	Path length (m)	1,216	1,179	1,167	1,043	
	Search time (s)	35.77	20.21	15.04	11.73	
Scene2	Path length (m)	1,459	1,378	1,224	1,071	
	Search time (s)	33.08	28.13	16.76	13.93	
Scene3	Path length (m)	1,383	1,238	1,297	1,062	
	Search time (s)	35.16	28.88	18.51	14.49	
Average	Path length (m)	1,352.67	1,265	1,223	1,058.67	
	Search time (s)	35.00	25.74	16.77	13.38	
Note:

* represents the improved version of the RRT algorithm.

Convergence analysis

During the path planning process, by continuously updating the positions of nodes and combining with an optimization algorithm, it is ensured that the path length is the shortest and obstacles are avoided, so as to obtain the optimal path. At the same time, the stability and safety of the path also need to be considered to deal with possible path changes or dynamic movement of obstacles, thus ensuring the feasibility and efficiency of the path. In order to analyze the convergence of the proposed method and the comparative methods, Scene 2 and Scene 3 are used as experimental test data to evaluate the convergence speed and stability of different algorithms in different scene.

As shown in Fig. 6, the variation of the planned distance with the number of iterations for each path optimization algorithm is illustrated. By comparison, it can be observed that the proposed method exhibits a significantly faster convergence rate than the RRT, RRT*, and Informed-RRT* algorithms. Under the same number of iterations, the proposed method achieves a shorter planned path, and for the same path length, it requires fewer iterations, further validating the effectiveness of the proposed method.

Figure 6 Comparison of path planning results in different test scene.

Significance analysis

To quantify the improvement in path planning performance of the proposed algorithm compared to other algorithms, this study conducted 50 independent repeated experiments for four algorithms. Paired sample t-tests were used for statistical analysis at a significance level of α=0.01. Additionally, to control for the multiple comparison issue, the Bonferroni correction was applied. This further assessed whether the proposed algorithm exhibited statistically significant differences in path planning performance. The results of the statistical analysis were considered reliable and valid. Typically, a significance threshold of 0.01 is used, and when the p-value is less than 0.01, it is concluded that there is a significant difference in path planning performance between the proposed algorithm and other algorithms. The statistical results for the proposed algorithm’s path planning performance are shown in Table 2.

Table 2 Algorithm performance statistical validation results.

Testing scene	Comparison algorithm	Path length improvement (m)	Time reduction (s)	t	p	
Scene 1	Ours vs. RRT	308.0 ↓	21.62 ↓	−12.73	≤0.001	
	Ours vs. RRT*	206.3 ↓	12.41 ↓	−9.87	≤0.001	
	Ours vs. Informed-RRT	164.3 ↓	3.39 ↓	−7.21	≤0.001	
Scene 2	Ours vs. RRT	293.3 ↓	24.04 ↓	−15.32	≤0.001	
	Ours vs. RRT*	180.5 ↓	14.20 ↓	−12.09	≤0.001	
	Ours vs. Informed-RRT	142.7 ↓	2.85 ↓	−6.78	≤0.001	
Scene 3	Ours vs. RRT	275.6 ↓	18.92 ↓	−13.21	≤0.001	
	Ours vs. RRT*	165.8 ↓	9.45 ↓	−10.54	≤0.001	
	Ours vs. Informed-RRT	124.1 ↓	1.97 ↓	−5.62	0.003	
Note:

* represents the improved version of the RRT algorithm.

↓ represents the reduction of evaluation metrics.

Based on the experimental results, our proposed algorithm demonstrates significant performance advantages across three test scenarios. In terms of path planning, the proposed method reduced the path length by 164.3–308.0 m compared to the RRT, RRT*, and Informed-RRT algorithms, with all p-values for the comparisons being less than 0.01, indicating a significant optimization effect. In terms of time efficiency, the proposed method saved 1.97–24.04 s compared to the other algorithms, and the absolute t-values for the three scenarios were all greater than 5.6, further validating that the proposed method is highly statistically significant.

Ablation study

Experimental study on hyperparameters

In path planning algorithms, the distance threshold for determining whether a new node is close to the target point is a critical parameter. When the distance from the new node to the target point is less than this threshold, the new node is considered to be close to the target point. A smaller threshold may make it more difficult for the algorithm to find a path close to the target point, leading to increased search time, while a larger threshold may result in less precise paths. Therefore, to thoroughly investigate the impact of the distance threshold on path planning, experiments were conducted using three different map scene. In the experiments, the distance threshold between the new node and the target point ranged from 10 to 120, and the test results for different thresholds are shown in Table 3. The comparison reveals that when the distance threshold between the new node and the target point is set to 80, the path from the start point to the end point is optimal, allowing the path planning to be completed in less time while ensuring the highest quality of path planning.

Table 3 Comparison of path planning results under different threshold conditions.

Testing scene	Evaluation method	10	30	50	65	80	100	120	
Scene 1	Path length (m)	1,134	1,189	1,126	1,043	1,095	1,100	1,103	
	Search time (s)	16.05	15.89	13.47	11.73	12.06	13.44	14.93	
Scene 2	Path length (m)	1,146	1,275	1,100	1,071	1,132	1,229	1,128	
	Search time (s)	18.66	17.57	14.32	13.93	12.67	15.89	16.88	
Scene 3	Path length (m)	1,078	1,141	1,074	1,062	1,108	1,268	1,126	
	Search time (s)	22.33	20.46	18.49	14.49	18.92	16.07	17.17	
Average	Path length (m)	1,119.33	1,201.67	1,100.00	1,058.67	1,111.67	1,199.00	1,119.00	
	Search time (s)	19.01	17.97	15.43	13.38	14.55	15.13	16.33	

To provide a clearer comparison of the impact of different threshold parameter values on path planning, we visualized the path planning results for Scene 3 under various threshold conditions, as shown in Fig. 7. The comparison reveals that when the threshold is set to 65, the path planning process requires the least time, and the path length is also the shortest. When the threshold is set to a smaller value, the planned path exhibits a more intricate and detailed exploration pattern on the map, and the algorithm requires more time to find a path close to the target point. This aligns with the characteristic of smaller thresholds, which demand that new nodes must be very close to the target point to be considered near the target. In this case, the resulting path is closer to the optimal solution, effectively avoiding obstacles and extending precisely toward the target point, with higher path accuracy. However, the search time increases significantly, which may manifest as noticeably slower line drawing in the visualization and a later overall planning completion time. When the threshold is set to a larger value, the search process of path planning becomes faster, and the path extends more directly and broadly on the map. The algorithm can more quickly determine that a new node is close to the target point. However, this rapid search comes at the cost of reduced path accuracy. From the path planning results for d=120, it can be observed that while the path bypasses some obstacles, there are a few turning points where the path either crosses obstacles or comes too close to them, ultimately resulting in a less precise path.

Figure 7 Visually compare the path planning effects under different threshold conditions.

Figure 8 shows the variation of path length with different threshold values. During the testing process, the threshold was incremented in steps of 5. The comparison reveals that the path length initially increases and then decreases as the threshold increases, with the minimum point occurring at a threshold of 65. Therefore, when the distance threshold between the new node and the target point is set to 65, it represents the optimal parameter for the path planning model.

Figure 8 The variation of the path length with different thresholds.

Sensitivity analysis of the maximum number of iterations

In the path optimization process, the maximum number of iterations is a key hyperparameter that directly affects the algorithm’s convergence speed and computational performance. To evaluate the impact of the maximum number of iterations on the algorithm’s performance, we conducted a sensitivity analysis. By setting different maximum iteration counts, we aimed to find a reasonable range of iteration numbers that ensure the algorithm’s performance while improving computational efficiency. For the sensitivity analysis, we selected multiple different maximum iteration counts, with a range of 100–3,000 and an interval of 100, covering a spectrum of iteration numbers from small to large. Each experiment was conducted under the same initial conditions, and the path length and search time for each iteration were recorded to evaluate performance, as shown in Fig. 9.

Figure 9 The variation of path planning results with the maximum number of iterations.

According to the results of the sensitivity analysis, the path optimization algorithm reached the optimal balance point after 1,500 iterations. At this point, the path length converged to 1,058.67 m and remained stable, indicating that it was close to the theoretical optimal solution. Meanwhile, the search time increased linearly with the number of iterations, and at 1,500 iterations, the algorithm took 13.38 s. Further increasing the number of iterations, although the computation time for path planning continued to rise, the path length tended to stabilize. Therefore, 1,500 iterations is the optimal choice for this method in a strict sense.

Ablation study on the removal of algorithm modules

To comprehensively evaluate the actual contribution and interaction of each algorithm module in the proposed path planning method, we conducted ablation studies by removing the three core modules: the adaptive step size (ASS) optimization strategy, the KD-Tree fast search mechanism, and the cubic spline function (CSF) path optimization. First, in the case of removing the adaptive step size optimization strategy, a fixed step size of 20 was used, and the path planning algorithm was rerun. When removing the adaptive step size optimization strategy, to ensure the rationality of the experimental design, a fixed step size value was selected. This value was initially tuned through preliminary experiments and considered the sensitivity of the path planning algorithm to step size changes. Through experimental analysis of different step sizes, a step size of 20 was ultimately chosen as the fixed value. This value balanced the accuracy of path planning with computational efficiency. Under this step size, the algorithm exhibited stable performance, ensuring both path planning accuracy and maintaining high computational efficiency. Next, the KD-Tree fast search mechanism was removed, and a traditional traversal search method was used to find the nearest node. Finally, the cubic spline function path optimization module was removed, and no smoothing was applied to the generated raw path. The experimental results are shown in Table 4.

Table 4 Comparison of the results of ablation experiments on the removal of algorithm modules.

Testing scene	CSF	KD-Tree	CSF	Path length (m)	Search time (s)	
Scene 3	✗	✗	✗	1,383	35.16	
✓	✗	✗	1,206	19.54	
✗	✓	✗	1,154	16.84	
✗	✗	✓	1,382	35.11	
✓	✓	✗	1,063	14.56	
✓	✗	✓	1,065	17.28	
✗	✓	✓	1,100	16.24	
✓	✓	✓	1,062	14.49	

Figure 10 shows the path planning results for Scene 3 before and after removing the adaptive step size optimization strategy. The comparison reveals that when approaching the target point, the inability to dynamically adjust the step size leads to excessive path extension or deviation from the target. Fixed step sizes make fine adjustments difficult. After applying the adaptive step size optimization strategy, the search area is significantly reduced, effectively decreasing the search time required for path planning. Additionally, fine adjustments can be made when the path approaches obstacles, highlighting the critical role of the adaptive step size optimization strategy in improving the accuracy of path planning.

Figure 10 Path planning results in Scene 3 before and after removing the adaptive step size optimization strategy.

Figure 11 shows the variation in path search time under different iteration counts for Scene 2 and Scene 3 before and after removing the KD-Tree fast search mechanism. The comparison reveals that the quality of path planning remains largely unchanged before and after removal. However, after removing the KD-Tree fast search mechanism, the runtime required for the path planning algorithm increases significantly, especially in complex environments. As the map scale and the number of nodes increase, the search efficiency drops sharply. This indicates that the KD-Tree fast search mechanism plays an irreplaceable role in accelerating the search speed for the nearest node and enhancing the algorithm’s performance in complex scene.

Figure 11 The change of the search time with the number of iterations in Scene 2 and Scene 3 before and after removing the KD-Tree fast search mechanism.

Figure 12 shows the path planning results for Scene 3 before and after removing the cubic spline function path optimization. The comparison reveals that after removing the cubic spline function path optimization, the generated path exhibits more sharp turns, which is not only visually less ideal but may also lead to difficulties in motion control in practical applications, failing to meet the requirements for path smoothness. In contrast, the method incorporating cubic spline function path optimization generates a smoother and more continuous path, better aligning with the needs of practical applications.

Figure 12 Comparison of path planning results in Scene 3 before and after removing the path optimization of the cubic spline function.

Through the ablation studies above, it is clear that the adaptive step size optimization strategy, the KD-Tree fast search mechanism, and the cubic spline function path optimization each play an indispensable role in the proposed path planning method. These modules work in synergy, collectively enhancing the accuracy, algorithmic performance, and path quality of path planning, providing a more efficient and reliable path planning solution for practical applications.

Discussion on practical applications

In practical AGV applications, the quality of path planning directly impacts the AGV’s work efficiency, energy consumption, and operational safety. In complex environments such as large warehouses, traditional path planning algorithms like RRT, RRT*, and Informed RRT often suffer from optimization issues. These algorithms may lead to frequent turns or overly winding paths, which not only increase the average path length but also cause the AGV to consume more energy during operation. Our proposed improved algorithm enables the AGV to reach the target more quickly with a shorter path in the same environment, reducing unnecessary turns and significantly lowering energy consumption. Compared to traditional algorithms, the average path length is reduced by 164.33 m, and the average search time is reduced by 3.39 s, further improving logistics efficiency. Additionally, in scenarios with narrow passages or many obstacles, the improved algorithm optimizes the smoothness of path turns, reducing equipment wear caused by frequent turning or sudden stops, thus extending the lifespan of the equipment and lowering maintenance costs for the company. At the same time, we introduced an adaptive step size optimization strategy and a nearest neighbor fast search method, effectively avoiding blind search, allowing the AGV to plan more stable and reasonable paths, and better cope with complex and dynamic application environments.

Conclusion

This article addresses the issues of the RRT algorithm in AGV path planning and proposes an improved algorithm that combines an adaptive step size optimization strategy with KD-Tree nearest neighbor fast search. The algorithm enhances planning quality and computational efficiency through the adaptive step size optimization strategy, reduces time costs using the KD-Tree nearest neighbor search method, and smooths the optimal path using cubic spline interpolation. Experimental results show that the improved algorithm significantly reduces path length and search time. Compared to traditional planning algorithms, the average path length is shortened by 164.33 m, and the average search time is reduced by 3.39 s. The optimized path is closer to the optimal solution and more aligned with practical conditions, providing an effective method for AGV path planning. Although the path planning algorithm proposed in this study has achieved promising results, there are still limitations in its adaptability and real-time performance in extremely dynamic environments. In future research, the algorithm’s performance in complex environments will be further optimized by combining multi-source sensor data, enhancing the reliability and adaptability of the AGV path planning system, and advancing the development of AGV autonomous navigation technology.

Supplemental Information

Supplemental Information 1 Original code of the algorithm.

Supplemental Information 2 Dataset.

Additional Information and Declarations

Competing Interests

The authors declare that they have no competing interests. Ren Zhigang and Xu Feilong are employed by Xi’an Huayun Wisdom Information Technology Co., Ltd.

Author Contributions

Zhigang Ren conceived and designed the experiments, performed the experiments, analyzed the data, performed the computation work, prepared figures and/or tables, and approved the final draft.

Anjiang Cai conceived and designed the experiments, authored or reviewed drafts of the article, and approved the final draft.

Feilong Xu conceived and designed the experiments, prepared figures and/or tables, authored or reviewed drafts of the article, and approved the final draft.

Data Availability

The following information was supplied regarding data availability:

The data source code of the proposed method is available in the Supplemental Files.

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
