# Peer review of "Automated guided vehicle (AGV) path optimization method based on improved rapidly-exploring random trees"

_PeerJ Computer Science, doi:10.7717/peerj-cs.2915_

## Round 0.1 · original submission · Major Revisions

In the opinions of three reviewers and mine, this paper can get a major revision.

Reviewer 1 ·

Basic reporting

The Article is well structured and well defined, but if you want to make it more effective and unique. So, the following we have some good suggestions and questions to improve. Each suggestion highlights a valuable aspect that, if addressed, could improve the article's effectiveness. For your knowledge, the mentioned comments will be a proper check after your revision.’’

Experimental design

The abstract effectively summarizes the research, but should explicitly state the key numerical improvements achieved by the proposed method. The abstract mentions "significant enhancements in path length, planning time, and path smoothness" but lacks precise numerical comparisons.

 The introduction effectively presents the background and motivation, but lacks a well-defined research gap. It discusses the limitations of existing RRT algorithms but does not clearly identify gaps in prior studies. Add a paragraph explicitly summarizing the gap and how the proposed method fills it.

 The methodology is well-structured but lacks a clear explanation of parameter selection. The adaptive step size optimization is discussed, but there is no explanation of how step sizes are determined or adjusted dynamically. Include a subsection detailing parameter tuning and sensitivity analysis.

Validity of the findings

The results provide comparative analysis, but statistical validation is missing. Performance improvements are mentioned, but significance tests (e.g., t-tests) are absent. Include statistical validation of improvements to strengthen claims. The discussion lacks a comparison with real-world AGV applications. Add a discussion on the practical applicability of the method.

 The conclusion summarizes findings well but does not mention limitations or future work. Add a brief section outlining potential improvements and real-world implementation challenges.

Additional comments

Suggestions for Improvement
 Ensure that parameter choices and algorithmic decisions are well-justified.
 Include statistical validation of improvements.
 Discuss the practical implications of the research beyond simulations.
 Improve readability by restructuring some technical explanations.

Reviewer 2 ·

Basic reporting

-

Experimental design

-

Validity of the findings

-

Additional comments

The manuscript presents an improved Rapidly-exploring Random Tree (RRT) algorithm for path planning of Automated Guided Vehicles (AGVs), addressing common limitations of traditional RRT methods such as low computational efficiency, slow convergence, tortuous paths, and susceptibility to local optima. The proposed method integrates an adaptive step size optimization strategy, a KD-Tree-based fast nearest neighbor search, and cubic spline interpolation for path smoothing. Comparative experiments with traditional RRT, RRT*, and Informed-RRT* are conducted to validate the approach. The topic is timely and relevant to industrial automation and AGV navigation, aligning well with the scope of PeerJ Computer Science. However, while the work shows promise, there are several areas where clarity and novelty could be enhanced to strengthen its contribution.
1) While the proposed improvements (adaptive step size, KD-Tree, cubic spline smoothing) are sensible, they are not entirely novel individually. Adaptive step sizes have been explored in prior RRT variants (e.g., Wang et al., 2023, cited in the paper), KD-Tree has been used for nearest neighbor searches in path planning (e.g., Guo et al., 2023), and cubic spline smoothing is a well-established technique. The novelty lies in their combination, but this is not sufficiently emphasized or justified.
2) The adaptive step size strategy (Equations 7-8) lacks detail on how thresholds dmin, dmax, ∇min, and ∇max are determined. Are these empirically tuned, analytically derived, or environment- specific? Their sensitivity is partially addressed in the ablation study, but the initial selection process is unclear.
3) The KD-Tree implementation is described generically, but its specific adaptation to AGV path planning (e.g., handling dynamic obstacles or high-dimensional spaces) is not detailed.

The computational complexity of KD-Tree construction and search is not analyzed, which is critical for efficiency claims.
4) The ablation study is a strength, but choosing a fixed step size of 20 when removing the adaptive step size module is arbitrary and unjustified.
5) The literature review is broad but lacks critical analysis of how cited works compare to the proposed method (e.g., how does RRT*-ACO by Wang et al., 2024, differ in approach?).
6) The conclusion summarizes the work but does not discuss limitations or future directions, limiting its forward-looking value.
7) There are some statements without reference. For example, “It is particularly adept at quickly exploring the entire search space in narrow paths, making it widely applicable in AGV path planning. However, the paths generated by the RRT algorithm can be somewhat random and may not be optimal, with pathfinding efficiency limited in environments with large-scale dense obstacles, requiring numerous iterations to successfully find a path.”

·

Basic reporting

The authors introduced an improved Rapidly-exploring Random Tree (RRT) algorithm that integrates adaptive step size optimization with KD-Tree fast matching search. To demonstrate the feasibility of the proposed algorithm, the authors have compared the improved RRT with traditional RRT, RRT*, and Informed-RRT*.

The idea of the proposed algorithm is very interesting. However, it is recommended to improve the quality of the paper based on the following suggestions and comments.

Experimental design

Pont 1: Section 1 and 2 are fine. However, this paper has a lack of references. Therefore, the authors are suggested to add some recent studies related to, e.g. decision tree algorithm, . For instance, (1) Density gradient-RRT: An improved rapidly exploring random tree algorithm for UAV path planning; (2) Rapidly-exploring Random Trees multi-robot map exploration under optimization framework; (3) More Quickly-RRT*: Improved Quick Rapidly-exploring Random Tree Star algorithm based on optimized sampling point with better initial solution and convergence rate; (4) RDT-RRT: Real-time double-tree rapidly-exploring random tree path planning for autonomous vehicles; (5) Bidirectional rapidly exploring random tree path planning algorithm based on adaptive strategies and artificial potential fields; (6) A rapidly-exploring random trees approach to combined task and motion planning; and (7) CDRT-RRT*: Real-time rapidly exploring Random Tree Star based on convex dissection. Please consider to comment and cite them in Section 2!

Point 2: The organization of this paper should be added in the last paragraph of Section 1.

Point 3: The pseudo-code of the proposed algorithm should be added in this paper.

Point 4: Please include runtime complexity of the proposed algorithm!

Point 5: What are the merits and limitations of the proposed algorithm?

Validity of the findings

In the las section, the authors stated that experimental results show that the improved algorithm significantly reduces both path length and search time. Compared to traditional planning algorithms, the average path length is shortened by 164.33 units, and the average search time is reduced by 3.39 seconds. This depends on the spesification of PC used in this study. So, I would like to suggest the authors to discuss runtime complexity of the proposed algorithm and other comparative algorithms in Section ABLATION STUDY.

Additional comments

Please make sure all equation, figures, and tables numbers have been cited in the main text!

---

## Round 0.2 · accepted · Accept

In the opinions of original reviewers and mine, this revised paper is able to be accepted now.

Reviewer 1 ·

Basic reporting

The manuscript is clearly written in professional and unambiguous English. The authors have provided a sufficient literature review and established a clear background and context for the study. The article follows a professional structure, with appropriate use of figures and tables to support the content. Raw data and relevant results are presented in a self-contained manner and aligned with the stated hypotheses. All technical terms are clearly defined, and the formal results are supported by detailed and valid proofs. I am satisfied with the authors' revisions and believe the manuscript meets the required standards for basic reporting.

Experimental design

The manuscript presents original primary research that aligns well with the Aims and Scope of the journal. The research question is clearly defined, relevant, and addresses a meaningful gap in the existing literature. The authors have conducted a rigorous investigation that adheres to high technical and ethical standards. The methodology is described in sufficient detail to allow for replication by other researchers. Overall, the experimental design is sound and supports the objectives of the study effectively.

Validity of the findings

The findings presented in the manuscript are valid and supported by robust and statistically sound data. The authors have provided all necessary underlying data, which are well-controlled and appropriately analyzed. The conclusions are clearly stated, logically follow from the results, and remain focused on the original research question. The study encourages meaningful replication and contributes valuable insights to the literature. Overall, the validity of the findings is strong and well-aligned with the study's objectives.

Additional comments

The authors have addressed previous concerns effectively and made appropriate revisions to improve the clarity and quality of the manuscript. The study is well-structured, methodologically sound, and provides a meaningful contribution to the field. I have no further suggestions at this stage and recommend the manuscript for publication in its current form.

Reviewer 2 ·

Basic reporting

The authors have addressed all my comments.

Experimental design

.

Validity of the findings

.

Additional comments

.